# Correct Me If I'm Wrong: Language Guided Exploration for VLAs

Sunshine Jiang*, John Marangola*, David Zhang*,
Nitish Dashora*, Pulkit Agrawal*, Zhang-Wei Hong†

*Massachusetts Institute of Technology
†MIT-IBM Watson AI Lab

*Abstract*—Reinforcement learning (RL) is a promising framework for improving vision language action (VLA) models through interaction, but it often struggles when the initial policy repeatedly fails in the same way. Our key observation is that these models already offer a simple way for humans to change what the policy attempts through language. In real world robot training, humans often need to stay nearby to monitor rollouts, reset environments, and check progress. This makes prompt updates a natural and lightweight source of guidance. While watching the robot, a human can redirect exploration by simply revising the instruction, without providing low level actions or directly controlling the robot. We introduce language-guided exploration, an interactive post-training framework that uses human prompt updates to help VLAs collect more useful experience. On real world physical robots, our method significantly improves sample efficiency over standard RL, making robot learning more practical under limited data regime. We further show that interactive post training makes VLAs more responsive to prompt updates at test time, enabling users to guide robot behavior through natural language during execution.

## I. INTRODUCTION

Vision-language-action (VLA) models make language a practical interface for robot control: a single policy can condition on an instruction, perceive the scene, and produce actions for many manipulation tasks [17, 6, 1]. In most RL post-training pipelines, language is used only as a fixed task specification, which we call the *canonical instruction*: the instruction that defines the reward and is used at deployment. The robot receives this instruction, executes a rollout, receives reward, and repeats. When the initial policy is weak, this fixed-instruction loop can repeatedly elicit the same failure pattern, such as approaching the wrong object or stalling before a subgoal (Figure 1, top). Since RL improves only from the data it collects, repeatedly collecting similar failures provides little useful signal.

This bottleneck is especially important for real-world VLA learning, where rollouts require physical interaction, environment resets, and safety monitoring. A human operator is therefore often already present during training, watching the robot and assessing progress. This creates a simple but underused opportunity: instead of passively watching repeated failures, the operator can guide exploration through the same language interface that the VLA already uses.

This paper studies language as an exploration interface for VLA post-training. The idea is simple: use language to change

what the policy attempts, while leaving action generation to the VLA. For example, if the canonical instruction is "open the top drawer and put the bowl inside" and the robot stalls before opening the drawer, the operator may make the next step explicit with "pull open the top drawer"; if the robot later lifts the bowl too low to clear the drawer, the operator may say "move the bowl higher" (Figure 1, middle). The robot continues to act through its learned policy, conditioned on the updated instruction. Compared with action intervention through a SpaceMouse or other control device, language intervention is lightweight and semantic: it does not require low-level controls, but can still redirect the rollout toward more useful behavior.

Prior work has shown that natural language is an effective interface for correcting robot behavior during execution. Language feedback has been used to correct robot plans [11], enable online manipulation corrections in shared autonomy [3], teach visuomotor policies from verbal feedback [8], and improve long-horizon robot behavior through on-the-fly language corrections [12]. Recent hierarchical VLA systems also incorporate situated user feedback during execution [13]. Our work builds on the same observation that language is a natural correction interface, but studies a different role for it: guiding exploration during RL post-training, while the reward and deployment target remain the canonical instruction.

We instantiate this idea as Instruction-driven Exploration (IDE), illustrated end-to-end in Figure 1. The figure summarizes the complete method in three stages. Under the canonical instruction, the SFT policy fails (top). During data collection, each rollout starts from the canonical instruction, but the human updates it online as the robot executes, producing a successful trajectory under a sequence of intervened instructions (middle). The policy is then updated by RL, with reward still defined by the canonical instruction, so that intervention shapes the experience collected without changing the task being optimized. After this update, the policy succeeds under the canonical instruction alone, with no intervention at test time (bottom).

We evaluate IDE on real-world physical robots. Compared with standard RL, IDE substantially improves sample efficiency under limited interaction. We further find that policies trained with language intervention become more responsive to natural language updates at test time, allowing users to guide execution when the robot starts to fail. These results suggest a simple shift in how language can be used in VLA post-training: beyond

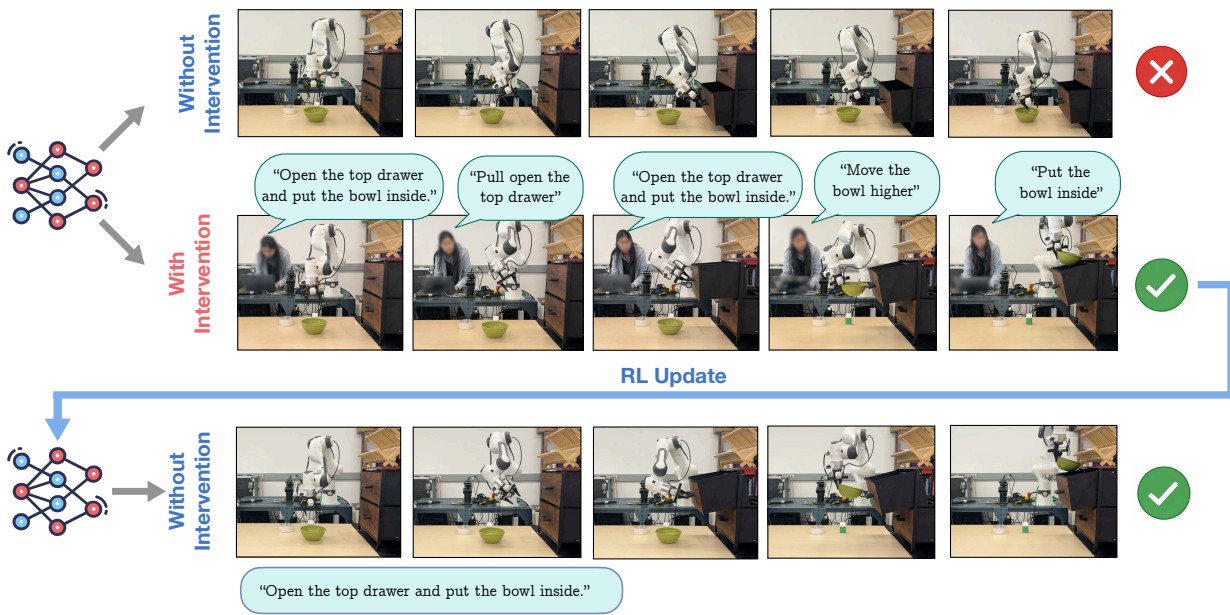

Fig. 1: **Instruction-Driven Exploration (IDE).** A human operator augments RL fine-tuning of a VLA policy with mid-rollout language corrections, transferring the corrected behavior back to the canonical prompt via policy gradient updates. *Top:* under the canonical prompt, the SFT policy fails. *Middle:* with mid-rollout language interventions, the policy completes each subtask and produces a successful rollout. *Bottom:* after RL fine-tuning on these instruction-driven rollouts, the policy succeeds under the canonical prompt alone, with no further intervention.

specifying the task, language can also provide an efficient interface for guiding exploration during learning.

## II. RELATED WORK

**Human-in-the-loop robot learning.** Human supervision has long been used to make robot learning more efficient through demonstrations, interventions, corrections, and expert labels [9, 14, 15]. These approaches are effective because humans can recognize failure modes and provide useful guidance. However, they often require low-level control, expert demonstrations, or action-level feedback. IDE is motivated by a lighter form of supervision that is natural for VLAs: when a human is already monitoring robot training, language is often the easiest way to redirect the robot without taking over control.

**Language feedback and robot correction.** Recent work has shown that natural language is an effective medium for correcting robot plans and behaviors. Language feedback has been used to modify planning objectives [11], support closed-loop embodied reasoning [4], teach policies from verbal corrections [8], and improve long-horizon robot behavior through on-the-fly language corrections [12]. Hierarchical VLA systems have also incorporated situated user feedback during execution [13]. These works show that language can correct or refine robot behavior. Our work studies a complementary role: using language feedback during RL data collection to steer exploration toward more useful experience.

**Prompting interfaces for robot control.** Several works study prompts as interfaces for robot policies, including multimodal prompts, visual prompts, marks, sketches, and interleaved image-text instructions [5, 7, 16]. These works show that the form of the prompt strongly affects grounding, planning, and control. IDE shares the view that prompts are active inputs that shape robot behavior, but focuses on a different question: how can prompt updates from a human monitoring the robot improve RL post-training? Our answer is to use language as an exploration interface, while using a mixed PPO update to prevent the policy from becoming dependent on corrected instructions at deployment.

## III. PRELIMINARIES

We consider RL fine-tuning for vision-language-action (VLA) policies [1, 6]. Each task is specified by a canonical language instruction $g \sim P_{\mathcal{G}}$, with $P_{\mathcal{G}}$ the task instruction distribution. The user provides $g$ at the start of a rollout. At each timestep $t$, the policy $\pi_\theta$ with parameters $\theta$ observes $o_t$ and samples $a_t \sim \pi_\theta(\cdot \mid o_t, g)$, continuing until the task is completed or the time limit is reached and producing a trajectory $\tau = (o_0, a_0, \ldots, o_{T-1}, a_{T-1}, o_T)$. As is common in manipulation, we use a binary reward $R(\tau, g)$ equal to 1 if $\tau$ completes $g$ and 0 otherwise, and maximize expected success $J(\theta) = \mathbb{E}_{g \sim P_{\mathcal{G}}, \tau \sim \pi_\theta(\cdot \mid g)}[R(\tau, g)]$. We optimize $J(\theta)$ with Proximal Policy Optimization (PPO) [10], using advantage

estimates $\hat{A}_t$ that measure how much better an action is than the policy's expected behavior at the same state. PPO minimizes the clipped surrogate loss

$$\mathcal{L}^{\text{PPO}}(\theta) = -\mathbb{E}_t\Big[\min\big(w_t(\theta)\hat{A}_t,\ \text{clip}(w_t(\theta), 1-\epsilon, 1+\epsilon)\hat{A}_t\big)\Big],$$
$$w_t(\theta) = \frac{\pi_\theta(a_t \mid o_t, g)}{\pi_{\text{old}}(a_t \mid o_t, g)}, \tag{1}$$

where $\pi_{\text{old}}$ is the policy that collected the rollout before the current update. Because the data come from $\pi_{\text{old}}$ but are optimized for $\pi_\theta$, the ratio $w_t(\theta)$ corrects for this mismatch and estimates how the update changes the likelihood of the sampled actions.

## IV. LANGUAGE AS AN EXPLORATION INTERFACE

**Problem.** Standard RL post-training holds the prompt $g$ fixed during data collection, sampling $a_t \sim \pi_\theta(\cdot \mid o_t, g)$ at every timestep. A weak initial policy driven by a single prompt tends to repeat the same failure, such as approaching the wrong object or stopping before contact, and these repeated failures carry little new information for policy improvement. In real-world fine-tuning each such rollout still costs robot time, resets, and human monitoring.

**Approach.** We introduce *Instruction-Driven Exploration* (IDE), which redirects data collection during RL through language intervention. This is practical in real-world fine-tuning because a human is typically already present to ensure safety and reset the environment. When the policy fails in an identifiable way, the human supplies a short correction to the prompt rather than taking over the robot's motions as in action intervention through a SpaceMouse. The VLA continues to generate all actions; only its language input changes.

**RL training with language intervention.** IDE updates the prompt online during rollout collection. Each rollout begins with the canonical prompt $g$ and sets the current prompt $\tilde{g} = g$. Let $\tau_{0:t} = \{(o_k, \tilde{g}_k, a_k)\}_{k=0}^{t-1}$ denote the partial trajectory before timestep $t$. At timestep $t$ the human observes $\tau_{0:t}$ and may replace the current prompt,

$$\tilde{g} \leftarrow \text{Human}(g, \tilde{g}, \tau_{0:t}),$$

for instance clarifying the target object after the robot reaches for the wrong one. The policy then continues from the updated prompt, $a_t \sim \pi_\theta(\cdot \mid o_t, \tilde{g})$.

The completed trajectory is scored against the canonical prompt through $R(\tau, g)$, even when parts of it were generated under intervened prompts. This keeps the learning objective tied to the original task while letting intervention shape exploration, and it raises two implementation challenges.

- **Prompt-distribution mismatch.** Data is collected under intervened prompts $\tilde{g}_t$, but deployment uses the canonical prompt $g$. Optimizing only under $\tilde{g}_t$ can improve the policy when a correction is present yet leave it unchanged under $g$.
- **Intervention timing.** Intervening too often makes training overly guided, while intervening too late wastes rollouts on repeated failures, so the procedure needs a way to decide when to intervene.

---

**Algorithm 1** Instruction-Driven Exploration (IDE)

---

**Input:** VLA policy $\pi_\theta$; canonical prompt $g$; human monitor Human; horizon $T$

1: $\tau \leftarrow \emptyset$, $\quad \tilde{g} \leftarrow g$
2: **for** $t = 0, \ldots, T-1$ **do**
3: $\quad$ Observe image $o_t$ and partial trajectory $\tau_{0:t}$
4: $\quad$ **if** intervention is allowed and the human chooses to intervene **then**
5: $\quad\quad \tilde{g} \leftarrow \text{Human}(g, \tilde{g}, \tau_{0:t})$ $\quad \triangleright$ update the prompt, not the action
6: $\quad$ **end if**
7: $\quad$ Sample $a_t \sim \pi_\theta(\cdot \mid o_t, \tilde{g})$; execute $a_t$; append $(o_t, \tilde{g}, a_t)$ to $\tau$
8: **end for**
9: Evaluate $R(\tau, g)$ using the canonical prompt and use $\tau$ for policy optimization

---

Algorithm 1 summarizes rollout collection. We address these two challenges through mixed backpropagation in PPO and adaptive intervention.

### A. Implementation

**Mixed backpropagation in PPO.** A rollout may be generated under a time-varying prompt sequence $\tilde{g}_{0:T-1}$ while the reward is evaluated under the canonical prompt $g$. For a transition $(o_t, \tilde{g}_t, a_t)$, the standard PPO ratio (Eq. III) updates the policy only under the prompt that generated the action, which does not guarantee success under $g$ at deployment. We instead average the current-policy log probability over the intervened and canonical prompts:

$$\ell_t(\theta) = \tfrac{1}{2}\log\pi_\theta(a_t \mid o_t, \tilde{g}_t) + \tfrac{1}{2}\log\pi_\theta(a_t \mid o_t, g), \tag{2}$$
$$w_t^{\text{mix}}(\theta) = \exp(\ell_t(\theta) - \log\pi_{\text{old}}(a_t \mid o_t, \tilde{g}_t)), \tag{3}$$

and use $w_t^{\text{mix}}(\theta)$ in the clipped objective. When $\tilde{g}_t = g$ this recovers standard PPO; when the human intervenes, the update raises the action likelihood under both the intervened and canonical prompts, so exploration improves while the learned policy stays aligned with the deployment prompt.

**Adaptive intervention.** We allocate intervention by the policy's recent success rate $\hat{s}(g)$ on prompt $g$, evaluated under the canonical prompt. The fraction of rollouts eligible for intervention is $\rho(g) = \rho_{\min} + (\rho_{\max} - \rho_{\min})(1 - \hat{s}(g))$, so low-success tasks receive more guidance and intervention decreases as the policy becomes reliable. Within an eligible rollout the human may update the prompt at any timestep $t$.

## V. EXPERIMENTS

We organize the experiments around four questions: (1) Can language intervention alone correct the failure modes of a frozen SFT policy, per task and per failure-mode subcategory (Section V-B)? (2) Does combining intervention with RL improve sample efficiency, wall-clock time, and human effort on real-world manipulation tasks (Section V-C)? (3) At test time, is the trained policy steerable mid-rollout, robust to

OOD object positions, and capable of zero-shot task transfer (Section V-D)? (4) Which design choices matter (Section V-E)?

*A. Setup*

**Task suite and platform.** We evaluate the effectiveness and efficiency of IDE for fine-tuning VLA models in the real world across four diverse manipulation tasks. These tasks span short-horizon object placement, compositional generalization to unseen object–container pairings, and long-horizon multi-stage manipulation. We use $\pi_{0.5}$ [2] as the base policy on a 7-DoF Franka FR3 with an AgileX gripper, operating over a tabletop scene with manipulable objects (tape, donut, bowl, green container), two drawers, and a two-level rack. RL training starts from a $\pi_{0.5}$ SFT checkpoint trained on the 10-task suite detailed in Appendix C.

**Tasks.** We evaluate on four tasks on which the SFT base policy is weak (success rate 0–20%) but whose failures are *correctable through language* rather than indicative of missing motor skills: in each case the policy already possesses the relevant primitives but misapplies them due to ambiguous grounding, an unfamiliar goal predicate, an OOD scene or object position, or weak instruction following. The four tasks together span the main sources of distribution shift—a non-SFT task with an OOD object position (*put the tape in the bowl*), a non-SFT task requiring an unseen object–predicate pairing (*put the tape in front of the camera*), an OOD object (*put the donut in the bowl*), and a long-horizon SFT task whose sub-skills are unreliably composed (*open the top drawer and put the bowl inside*). Each failure is thus a grounding or composition gap rather than a motor deficit, ensuring that observed gains are attributable to the language intervention itself. Full per-task descriptions and the relationship of each task to the SFT distribution are given in Appendix D.

**Baselines.** We compare IDE against two baselines covering the natural alternatives of improving the policy without a human and with teleoperated human corrections:

- **Proximal Policy Optimization (PPO)** [10]: standard on-policy RL with stochastic action sampling, conditioning the policy on the canonical task prompt $p_g$ throughout training.
- **DAgger** [9]: interactive imitation learning in which a human operator provides teleoperated corrections at self-selected timesteps with a spacemouse, updating the policy via behavior cloning on the corrected trajectories.

All methods start from the same SFT checkpoint, share the same evaluation protocol, and receive the same rollout-or-wall-clock budget.

*B. What Can Language Intervention Steer in a Frozen Policy?*

For language intervention to improve the sample efficiency of RL, it must be able to steer the policy's behavior. Before coupling intervention with training, we ask whether language alone can already redirect the SFT checkpoint in useful ways without weight updates.

**Semantic steering.** Language resolves which object the instruction refers to and which sub-goal applies. This matters when the policy is biased by similar training tasks (e.g., "put

the green container in the bowl" pulls attention toward the green container for any bowl-placement task) or faces objects absent from training (e.g., *donut*, *banana*). A single correction that names the target unambiguously re-routes the policy onto the correct object or sub-goal (Fig. 2, top, purple bubbles).

**Motor control refinement.** With the target correctly identified, language also shapes the motor sequence itself, tightening grasps, adjusting contact points, and accounting for workspace geometry. Prompts such as "*from the top*" or "*move higher*" tighten approach angles and adjust execution height with no weight updates (Fig. 2, bottom). We attribute this to motion associations already learned during training (e.g., "put the bowl on the top rack" pairs "top" with upward motion), which lets language both *redirect* the policy among existing options and *refine* how each is executed.

Table I (left two columns) reports per-task success rates on the frozen SFT checkpoint with and without intervention. Without any weight updates, intervention improves success on every task, with the largest gains where the SFT checkpoint scores zero.

*C. How Learning Benefits from Language Intervention*

Section V-B showed that language intervention reliably produces successful rollouts on a frozen SFT policy. We now ask whether feeding those rollouts back into RL training yields a better policy. On the same four tasks, we compare IDE against the SFT initialization and the PPO and DAgger baselines (Section V-A).

**Higher final success rate.** IDE attains the best final success rate on every task, more than doubling DAgger's average, with the largest gap on the harder tasks (Table I). Because IDE rewards any successful trajectory rather than imitating the operator's specific actions, the policy can improve past the demonstrated behavior instead of cloning it. PPO does not reach DAgger's success rate in three iterations, as expected: it receives no human intervention and thus exploration is hard, whereas DAgger receives direct action corrections.

**Better sample efficiency.** IDE reaches these success rates with far fewer rollouts than PPO (Fig. 3). It matches DAgger within the first iteration and keeps improving, while PPO improves slowly or not at all, staying at zero. The drawer task makes the mechanism clear: the SFT checkpoint scores zero under $p_g$, so action-noise PPO collects no successful rollouts and the policy gradient never sees a positive signal. Intervention prompts ("*open the top drawer and put the tape inside*" to disambiguate the drawer, then "*move the bowl higher*" to avoid pushing it shut) turn this into many successes per batch before any weight update, so the first IDE update is already well-conditioned.

**Shorter episodes.** IDE also succeeds more efficiently: its successful rollouts are the fastest on all four tasks, beating both SFT and DAgger (Table II). DAgger is in fact slower than SFT on two tasks, consistent with it dragging the policy through the task via corrective actions rather than teaching a better strategy. Higher success combined with shorter episodes

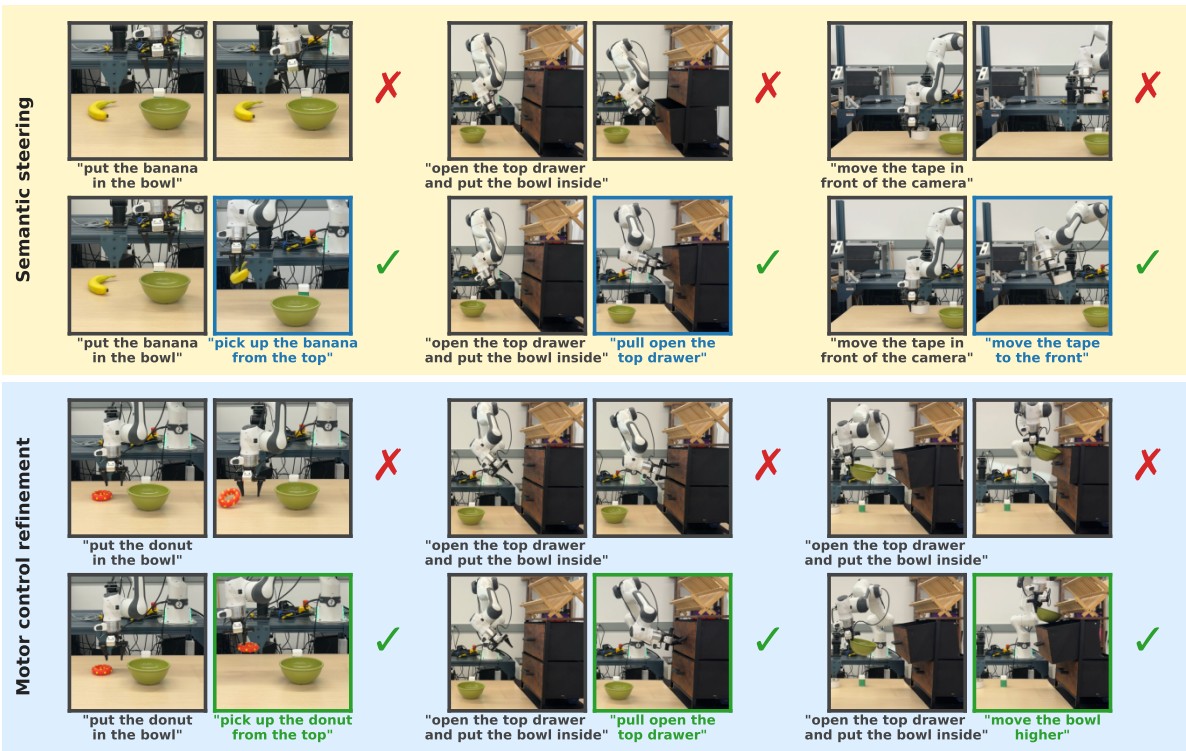

Fig. 2: **Language intervention steers a frozen SFT policy at both levels** (no weight updates). Each pair shows the canonical prompt failing (✗) and the intervention prompt succeeding (✓). **Top, Semantic steering:** the policy targets the wrong object or sub-goal; a blue prompt naming the target unambiguously redirects it. **Bottom, Motor control refinement:** the policy targets correctly but executes poorly; a green motor-cue prompt ("*from the top*", "*move the bowl higher*") tightens execution.

TABLE I: **Per-task success rate (%) on real-world tasks.** *SFT*: frozen checkpoint under the canonical prompt $p_g$; *SFT +p*: with an operator-chosen intervention prompt (Fig. 2), no weight updates in either case. *PPO*, *DAgger*, *IDE*: final success rates after RL fine-tuning, over 20 eval rollouts per task; (+X) is the gain in percentage points over SFT. Intervention alone improves every task; IDE builds on this through RL to attain the highest final success rate on every task.

| Task | SFT | SFT $+ p$ | PPO | DAgger [9] | PPO + IDE (Ours) |
|---|---|---|---|---|---|
| "put the tape in the bowl" | 20.0 | 70.0 (+50.0) | 30.0 (+10.0) | 30.0 (+10.0) | **75.0** (+55.0) |
| "put the tape in front of the camera" | 0.0 | 50.0 (+50.0) | 15.0 (+15.0) | 35.0 (+35.0) | **60.0** (+60.0) |
| "put the donut in the bowl" | 5.0 | 35.0 (+30.0) | 20.0 (+15.0) | 15.0 (+10.0) | **55.0** (+50.0) |
| "open the top drawer and put the bowl inside" | 0.0 | 10.0 (+10.0) | 0.0 (+0.0) | 10.0 (+10.0) | **35.0** (+35.0) |
| **Average** | 6.3 | 41.3 (+35.0) | 16.3 (+10.0) | 22.5 (+16.2) | **56.3** (+50.0) |

suggests IDE internalizes the corrected behavior rather than relying on prompt-time rescue.

**Less human effort.** IDE trains faster than DAgger on every task (Table II), with the largest gap on the drawer task, where teleoperating a precise handle grasp under arm occlusion is slow. PPO's runs are shorter only because failed rollouts terminate early, trading wall-clock for a much lower success rate. The decisive cost is human time: a typed prompt takes $\sim 2\,\mathrm{s}$ versus $\sim 30\,\mathrm{s}$ for a SpaceMouse correction, so the operator is in active control for only $\sim 5\%$ of training under IDE versus $\sim 75\%$ under DAgger.

*D. Does the Trained Policy Generalize Better and Remain More Steerable?*

Because IDE trains on a diverse distribution of intervention prompts, we expect two properties unavailable to single-prompt training. **(1) Generalization**: diverse prompts force the policy to ground language onto behavior rather than memorize one mapping, so it should handle unseen objects, scenes, and phrasings. **(2) Steerability**: at deployment, an operator should be able to recover a failing rollout with a language correction rather than restart the task.

We test both on the latest IDE, PPO, and DAgger checkpoints

TABLE II: **Training cost and rollout duration on real-world tasks.** *Training time* is the wall-clock cost of three iterations of 32 rollouts each, including operator intervention. *Success rollout duration* is the mean wall-clock of successful eval rollouts at the final iteration; (Nx) is the speedup over SFT, $N = $ SFT/method. The duration average covers the first three tasks only, as SFT and PPO have no successful rollouts on the drawer task. IDE matches PPO's training time, well below DAgger's, with the shortest successful rollouts.

| Task | Training time (min) | | | Success rollout duration (s) | | | |
| --- | --- | --- | --- | --- | --- | --- | --- |
| | PPO | DAgger [9] | IDE (Ours) | SFT | PPO | DAgger [9] | PPO + IDE (Ours) |
| "put the tape in the bowl" | 61.8 | 87.0 | 68.1 | 20.5 | 19.2 (1.1x) | 24.9 (0.8x) | **18.0** (1.1x) |
| "put the tape in front of the camera" | 29.2 | 62.0 | 44.9 | 19.9 | 15.3 (1.3x) | 11.5 (1.7x) | **10.8** (1.8x) |
| "put the donut in the bowl" | 66.4 | 85.0 | 60.2 | 24.7 | 29.3 (0.8x) | 37.3 (0.7x) | **25.8** (1.0x) |
| "open the top drawer and put the bowl inside" | 55.3 | 131.0 | 70.1 | — | — | 48.0 | **36.4** |
| **Average** | 53.2 | 91.3 | 60.8 | 21.7 | 21.3 (1.0x) | 24.6 (0.9x) | **18.2** (1.2x) |

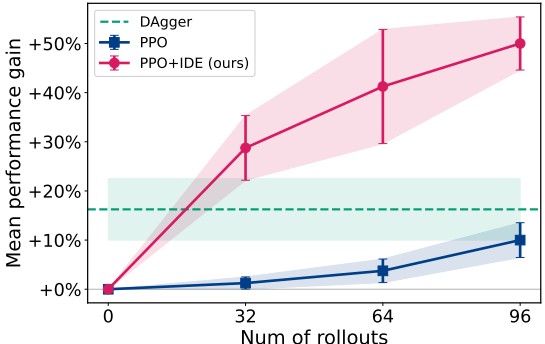

Fig. 3: **Performance gain over SFT vs. training rollouts**, averaged across four tasks. IDE surpasses DAgger within the first iteration and keeps improving; PPO stays near SFT. Bands: $\pm 1$ standard error across tasks.

TABLE III: **Test-time generalization and steerability on "*put the tape in the bowl*".** Success rate (%) over 10 rollouts per condition. "w/o" = no corrections (generalization); "w/" = operator may issue sub-skill prompts (generalization + steerability); $\Delta$ is the gain from correction.

| | PPO | | | DAgger | | | PPO + IDE (Ours) | | |
| --- | --- | --- | --- | --- | --- | --- | --- | --- | --- |
| | w/o | w/ | $\Delta$ | w/o | w/ | $\Delta$ | w/o | w/ | $\Delta$ |
| with distractor | 0 | 10 | +10 | 10 | 0 | −10 | 10 | **50** | **+40** |
| OOD object + pose | 0 | 0 | 0 | 0 | 0 | 0 | 0 | **10** | **+10** |

under two test-time shifts of "*put the tape in the bowl*": (i) a distractor (blue tape) in the scene, and (ii) an OOD object (banana) at an OOD position (Fig. 4). Each is run with and without mid-rollout corrections (Table III): the "w/o" columns isolate generalization under $p_g$, the $\Delta$ columns measure the steerability lift. IDE leads on both.

**IDE generalizes better without correction.** On the distractor shift, IDE ties DAgger and beats PPO, which is captured by the distractor and reaches for the wrong object (Table III, "w/o"). Under the joint OOD shift all three score zero, but the failures differ: PPO and DAgger route to the green container, while IDE occasionally picks up the banana without completing

the task. Diverse-prompt training thus yields a checkpoint that does not collapse under scene shift.

**IDE is the only checkpoint that benefits from mid-rollout correction.** Fig. 4 shows the divergence directly: given the same correction prompt at the same step, only IDE redirects to the banana and, with two further corrections, completes the deposit. The pattern holds quantitatively (Table III, $\Delta$): IDE gains $+40$ pp on the distractor shift and $+10$ pp on the OOD shift, PPO gains little, and DAgger *loses* 10 pp—reprompting actively destabilizes it. PPO's narrow training-prompt distribution leaves it unable to interpret sub-skill phrasings; when it does react, the prompt drives it toward an OOD end-effector pose, an artifact of overfitting. DAgger's recovery is bound to the in-distribution scene.

**Why IDE generalizes and steers better.** Both properties trace to training over joint prompt and scene variation. PPO sees one prompt per task and never grounds language variation in behavior. DAgger sees diverse *actions* in a fixed scene, tying its recovery to that scene rather than to language. IDE covers both, preserving the prompt-conditioning structure of the underlying VLA.

**Steerability is robust to operator choice.** The test-time steerability result could depend on the specific operator composing the corrections. We recruit three operators of varying robotics experience and have each issue mid-rollout corrections to the same final IDE checkpoint on the two shifts above (10 rollouts per task, no further training). Per-rollout success rate averages 46% across operators (with $\sigma = 0.04$) on the distractor shift, indicating the policy is not over-fit to any operator's style (Table IV). Operators converge on similar prompts under *object misidentification*, where phrasing is constrained to forms like "*grasp the X*", and diverge more under *missing sub-skills*, where several phrasings can recover the behavior.

*E. Which Design Choices Matter?*

**Backpropagation variant.** We compare three variants: *Orig* (backpropagate only through the canonical prompt $p_g$), *Mod* (only through the intervention prompt $p$), and *Mix* (through both; our default). Mix substantially outperforms both ablations (Fig. 5, left). Orig discards the intervention-prompt gradient— the very signal that lets IDE escape sparse reward. Mod trains

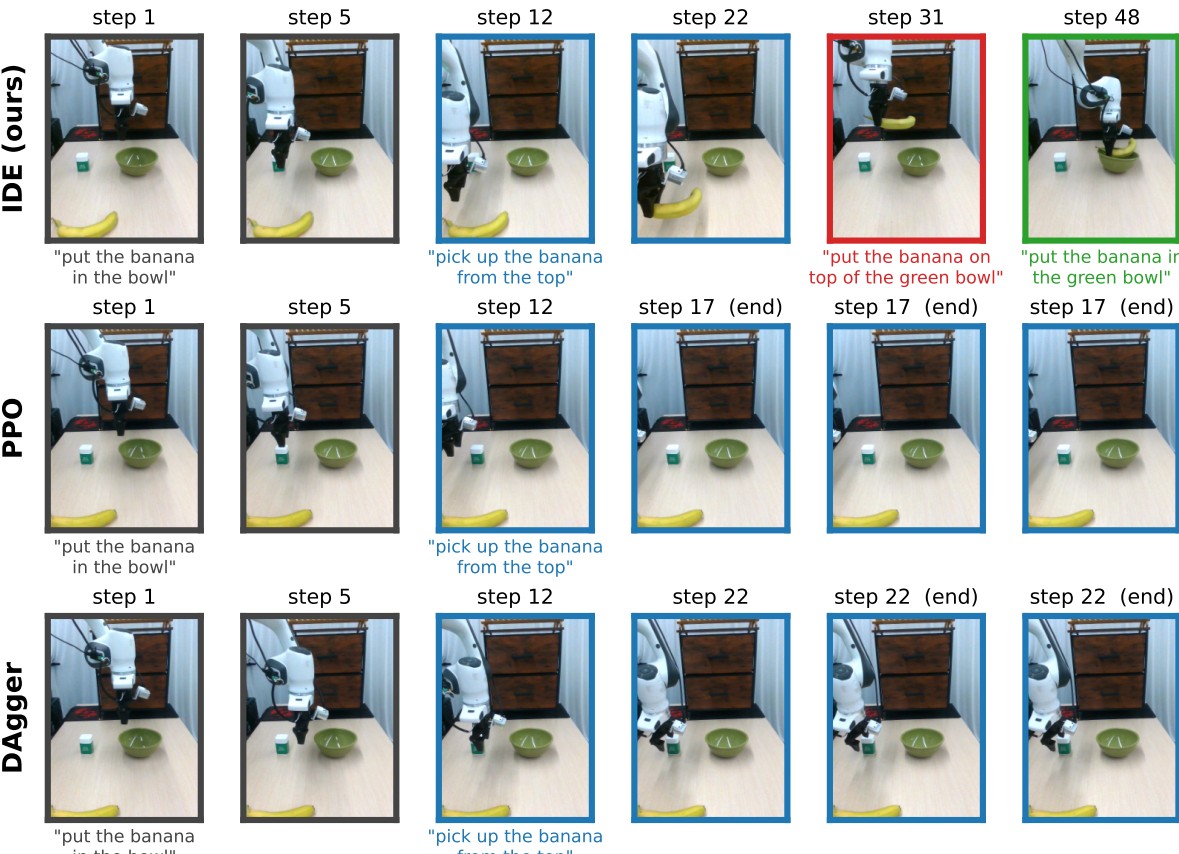

Fig. 4: **Same scene, same correction prompt, different responses.** Each row shows a rollout under $p_g$="*put the banana in the bowl*" (gray border) followed by the same mid-rollout correction "*pick up the banana from the top*" (blue border) at step 12. All three checkpoints initially approach the green container instead of the banana. **IDE (top):** the correction redirects the gripper to the banana; two further corrections complete the task. **PPO (middle):** the policy does not respond to the correction and terminates in an OOD pose at step 17. **DAgger (bottom):** the policy ignores the correction and continues toward the green container.

TABLE IV: **Per-operator test-time success rate on the same IDE checkpoint.** Each operator runs 10 rollouts per task with mid-rollout language corrections enabled; op1, op2, op3 are operators with increasing robotics experience. $\sigma$ is the standard deviation across operators.

| Task | op1 | op2 | op3 | avg | $\sigma$ |
|---|---|---|---|---|---|
| with distractor | 50% | 50% | 40% | 46% | 0.04 |

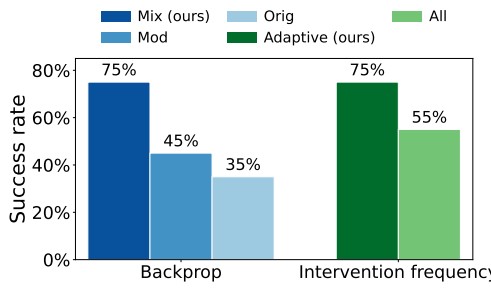

Fig. 5: **Ablations of IDE's two design choices on "*put the tape in the bowl*".** **Left:** backpropagation variant. *Mix* (ours) uses both prompts; *Mod* only the intervention prompt; *Orig* only the canonical prompt. **Right:** intervention frequency. *Adaptive* (ours) scales inversely with current success rate; *All* intervenes on every rollout. Both ablations underperform the defaults, confirming that both design choices are necessary.

the policy to succeed under intervention prompts but never under $p_g$, the prompt used at evaluation. Mix combines both signals, training the policy to execute the task under $p_g$ while preserving the responsiveness to correction prompts that makes the checkpoint steerable at test time (Section V-D).

**Intervention frequency.** We compare *All* (intervene on every rollout) against *Adaptive* (frequency inversely proportional to current success rate; our default). Adaptive outperforms All (Fig. 5, right). Intervening on every rollout exposes the policy almost only to corrected—and therefore successful—rollouts,

leaving it unable to recover from the failure modes that drove the interventions in the first place. Scaling corrections down with success rate restores this exposure: at low success rate, intervention dominates and the policy learns the corrected behaviors; at high success rate, the policy handles most rollouts on its own and learns to recover from intermediate failures.

*F. Qualitative Analysis*

**Emergent behaviors.** Inspecting the trained policies reveals two patterns, both of which expand what the policy can do beyond the SFT distribution. *New skills not present in the SFT data*: IDE-trained policies acquire behaviors that no demonstration ever shows. For example, on "*put the tape in the bowl*" all training demonstrations grasp the tape from the left side, but a correction prompt during training leads the policy to acquire both a right-side grasp and a matching placement strategy. The behavior persists at evaluation under the canonical prompt with no intervention. *Robustness to intermediate failures*: IDE-trained policies recover from intermediate mistakes instead of abandoning the task. They retry after dropping an object or missing a grasp; when the bowl gets knocked to the edge of the table during a placement attempt, they reposition and complete the placement; when the tape is dropped outside the bowl, they pick it back up and place it inside. On the drawer task, the upward-raising motion elicited by "*move the bowl higher*" is absorbed into canonical-prompt rollouts, so the bowl no longer collides with the drawer rim by later iterations.

**Operator–policy co-adaptation.** A distinctive feature of IDE training is that the operator's intervention vocabulary shifts as the policy improves and new failure modes surface. On the drawer task, the dominant failure in the first training iteration is missing the top drawer handle, so the operator's prompts mostly target drawer disambiguation and grasping; once the policy can reliably open the drawer, the dominant failure shifts to the bowl colliding with the drawer front, and the operator switches to prompts that lift the bowl higher. This keeps each training rollout focused on the policy's current dominant failure mode—a property static intervention protocols cannot offer, since the human is not running through a fixed correction script but responding to whatever the policy is currently getting wrong.

## VI. CONCLUSION

In this paper, we introduce IDE, which enables human operators to guide RL exploration through language interventions during training, substantially improving sample efficiency.

**Limitations.** The effectiveness of IDE depends on the language-conditioned controllability of the underlying VLA. We observed cases where language alone was insufficient: prompts could not reliably recover when the arm moved to the edge of the camera view, and object-name confusion sometimes required operators to use nearby reference objects or unnatural wording. Stronger language-grounded controllability would allow future VLAs to benefit more from IDE.

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

APPENDIX

*A. Hardware and control stack.*

Observations consist of two RGB streams (a wrist camera and a global side camera, both $320 \times 240$ at $60\,\mathrm{Hz}$) and the robot's 10-d proprioceptive state (3-d end-effector position, 6-d continuous rotation, 1-d gripper width). Actions are 10-d absolute end-effector targets dispatched to a $1\,\mathrm{kHz}$ operational-space impedance controller. The VLA produces chunks of 50 actions per inference call; we execute the first 16 before re-querying, yielding closed-loop control at $50\,\mathrm{Hz}$ with an effective re-planning rate of roughly $3\,\mathrm{Hz}$.

*B. Intervention and termination protocol.*

Each rollout is terminated and labelled online by a human operator who presses `s/f/r` for success, failure, or reset; this replaces a learned binary success classifier. For intervention rollouts, the operator may additionally press `p` to pause and swap the natural-language prompt mid-trajectory (e.g., switching from "put the tape in the bowl" to the recovery prompt "pick up the tape from the top").

*C. SFT Task Suite*

The $\pi_{0.5}$ checkpoint used as the starting point for all RL fine-tuning experiments is trained on the ten language-conditioned manipulation tasks listed in Table VI. The tasks share a fixed tabletop scene containing four manipulable objects (tape, bowl, green container, and a small object placed on a marked region of the workspace), two stacked drawers (top and bottom), and a two-level rack. Together they span four object–receptacle relationships—placing an object *in* a bowl (task 1), *on* a rack (tasks 2, 3, 10), *in front of* the camera (tasks 4, 5), and *inside* a drawer (tasks 6–9)—over three target objects (tape, bowl, green container) and four receptacle locations (bowl, top rack, bottom rack, top drawer, bottom drawer, front of the camera).

Each task is collected via teleoperated demonstrations on the Franka FR3 platform described in Section V-A, with 40 demonstrations per task and object positions randomized within a fixed workspace region across demonstrations. We hold the demonstration budget fixed across tasks rather than scaling to per-task difficulty, so the SFT checkpoint's per-task competence reflects the intrinsic difficulty of each task under uniform supervision.

*D. Evaluation Task Details*

The four evaluation tasks used in the main experiments are constructed around the SFT task distribution described in Appendix C. Each task probes a distinct axis of OOD-ness while keeping the underlying motor skills within reach of the SFT-trained primitives, so that observed performance gains can be attributed to the language-grounding mechanism rather than to motor-skill acquisition. Per-task descriptions follow.

- **Put the tape in the bowl**: not in the SFT distribution, with the tape in an OOD position. The grasp and place primitives are known; language must recombine them under a new task structure and redirect attention to the displaced object.
- **Put the tape in front of the camera**: not in the SFT distribution. The tape is a seen object, but the goal predicate "in front of the camera" has never been paired with it; language must bind a seen object to a seen-but-uncombined predicate.
- **Put the donut in the bowl**: the donut is an OOD object unseen during SFT. The placement skill is familiar; language must transfer it to a new object identity.
- **Open the top drawer and put the bowl inside**: a long-horizon task that *is* one of the 10 SFT tasks but on which the checkpoint is weak. Language must reliably sequence sub-skills the policy can already execute individually.

We provide additional analyses that complement the main text. Section E compares IDE to DAgger from the on-policyness perspective, explaining why DAgger's teleoperated corrections can leave the policy in states it does not know how to recover from.

*E. On-Policyness vs. DAgger*

DAgger and IDE both inject a human correctional signal into RL fine-tuning, but they do so in fundamentally different ways. In DAgger, the human takes physical control and teleoperates a corrective segment; the actions in those segments are drawn from the human's behavior distribution, not from $\pi_\theta$. IDE, by contrast, never relinquishes control of the end-effector. The human modifies the prompt, but every action in every collected trajectory—including those collected under intervention prompts—is sampled from $\pi_\theta$ itself.

**Why does this matter?** When a DAgger operator teleoperates a correction, they often place the end-effector in configurations the policy has never seen. On put the tape in the bowl task, the human frequently drives the gripper directly onto the tape and triggers a grasp. From the policy's perspective, this is an OOD state: the gripper is near a graspable object in a configuration not present in training. When the rollout resumes under policy control, the policy often opens the gripper and moves away, undoing the correction. IDE avoids this by construction: because all actions are produced by $\pi_\theta$, the end-effector only ever visits states reachable under the current policy.

TABLE V: **Per-task failure-mode catalog and example intervention prompts.** (A) = scene/task confusion; (B) = action-execution error. Intervention prompts re-target the policy onto the correct object or invoke a related sub-skill from the SFT distribution.

| Task | Cat. | Observed failure mode | Intervention prompt |
|---|---|---|---|
| "put the tape in the bowl" | A | Goes to the green container or bowl (bias from "put the green container in the bowl" in SFT) | "Pick up the tape from the top" (avoids mentioning bowl/container) |
| | B | Overshoots the tape's OOD position (returns to SFT tape position) | "...from the top" biases approach angle and reduces overshoot |
| | B | Right-side grasp prevents placement in the bowl | The correction motion after right-side grasp is learned with RL training |
| "open the top drawer and put the bowl inside" | A | Goes to bottom drawer, or oscillates between top and bottom (bias from "open the bottom drawer" SFT task) | "Pull open the top drawer" (disambiguates which drawer to open) |
| | B | Reaches the top drawer handle but misses the grip | "Pull open the top drawer" (tightens the approach to the handle) |
| | B | After opening, pushes the drawer shut while still holding the bowl | "Move the bowl higher" (invokes upward motion that clears the drawer rim) |
| "put the donut in the bowl" | A | Hesitates between donut and bowl; sometimes picks up the bowl first; sometimes goes to the green container | "Pick up the donut from above" (names the target and the approach direction unambiguously) |
| | B | Imprecise grasp on the donut; pushes it to an OOD position; sometimes fails to release | "Pick up the donut from the top" (tightens the approach to a reliable top-down grasp) |
| "put the tape in front of the camera" | A | After picking up the tape, goes to the drawer or top rack (bias from "open the top drawer and put the tape inside" and "put the tape on the top rack") | "Move the tape to the front" (re-targets to the goal predicate) |
| | B | Does not reach the tape; misses the grasp | "Pick up the tape from the top" |

TABLE VI: **SFT task suite.** The ten tasks the $\pi_{0.5}$ SFT checkpoint is trained on, with 40 teleoperated demonstrations per task. The four evaluation tasks used in the main experiments are constructed around this distribution; see Section V-A.

| # | Prompt |
|---|---|
| 1 | *"put the green container in the bowl"* |
| 2 | *"put the tape on the top rack"* |
| 3 | *"put the bowl on the top rack"* |
| 4 | *"put the green container in front of the camera"* |
| 5 | *"put the bowl in front of the camera"* |
| 6 | *"open the top drawer and put the bowl inside"* |
| 7 | *"open the bottom drawer and put the bowl inside"* |
| 8 | *"open the top drawer and put the tape inside"* |
| 9 | *"open the top drawer and put the green container inside"* |
| 10 | *"put the green container on the bottom rack"* |