# OpenReview forum: "Correct Me If I’m Wrong: Language Guided Exploration for VLAs"
_roboticsfoundation.org/RSS/2026/Workshop/RL4VLA — RL4VLA_

### Official Review · Reviewer_azSa · 2026-06-28
**good paper with simple but effective idea of human-in-the-loop RL for VLAs**

**Rating:** 7
**Confidence:** 4

**Review:**

This paper addresses inefficient real-world RL fine-tuning of VLA models via prompting with language corrections by humans. A mixed PPO update addresses the prompt-distribution mismatch between canonical prompts and intervened ones, which transfers corrected behaviour back to deployment-time prompting.

This paper is clear, well-organised, and highly relevant to language-guided robot RL learning.  The real-robot experiments against PPO and DAgger make it practically compelling. However, the intervention mechanism remains slightly under-specified given its importance. It is unclear how intervention timing is decided, what qualifies as an identifiable failure, how often corrections are needed, and how robust the method is across operators. The approach may also depend heavily on failures being language-correctable rather than requiring new motor skills.

---

### Official Review · Reviewer_pv8e · 2026-06-28
**Review of paper "Correct Me If I’m Wrong: Language Guided Exploration for VLAs"**

**Rating:** 6
**Confidence:** 4

**Review:**

Summary

The paper presents IDE, a human‑in‑the‑loop RL method that uses online language interventions to guide exploration while preserving the canonical task objective: each rollout begins with a canonical instruction, humans edit that instruction in real time to produce successful trajectories, and the policy is updated with rewards defined by the original canonical instruction so the learned behavior succeeds without intervention at test time. Evaluated on four real‑world manipulation tasks—short‑horizon placement, compositional generalization to unseen object–container pairs, and long‑horizon multistage manipulation, it shows IDE substantially improves sample efficiency over PPO and DAgger when starting from the same SFT checkpoint and budget and yields policies that are more responsive to natural language at test time.

Strengths
1. This work explores RL fine‑tuning of multimodal foundation models for robotics, which is strong alignment with workshop themes.
2. Experiments on four diverse manipulation tasks (short‑horizon placement, compositional generalization, long‑horizon multistage) demonstrate practical utility and sim‑to‑real relevance.
3. The backpropagation variants (Orig, Mod, Mix) and intervention frequency (All vs Adaptive) directly address some design choices; the finding that Mix and Adaptive perform best is informative.
4. Policies trained with interventions become more responsive to natural language updates at test time, which is a valuable secondary benefit.

Weaknesses
1. While two ablations are present, the paper could benefit from analyzing sensitivity to learning rates for prompt components, intervention length/content, and timing of interventions within rollouts.
2. There are no experiments showing behavior under noisy, inconsistent, or adversarial interventions, nor explicit safety constraints during human‑guided exploration. Real‑world deployment requires these analyses.
3. Tasks are all manipulation; scalability to higher‑dimensional control, multi‑agent settings, or tasks with complex contact dynamics is not demonstrated.

Questions for the Authors
1. What instructions or constraints were given to human operators (vocabulary, length limits, templates)? Were interventions freeform or guided?

---

### Decision · Program_Chairs · 2026-07-03

**Decision:**

Accept

**Comment:**

This paper presents a human-in-the-loop RL framework that uses language interventions to improve real-world robot learning. The reviewers found the problem important and liked the real-world results. The main concerns are about the clarity of the intervention process and the discussion of the method's limitations. Overall, we believe the paper is a valuable contribution to the workshop. For the camera-ready version, the authors should better describe the intervention protocol.